# Genetic Variation of *Magnaporthe oryzae* Population in Hunan Province

**DOI:** 10.3390/jof9070776

**Published:** 2023-07-23

**Authors:** Zhirong Peng, Yuefeng Fu, Fan Wang, Qiqi Liu, Yi Li, Zhengbing Zhang, Li Yin, Xiao-Lin Chen, Jingbo Xu, Huafeng Deng, Junjie Xing

**Affiliations:** 1State Key Laboratory of Hybrid Rice, Hunan Hybrid Rice Research Center, Changsha 410125, China; 2Yueyang Academy of Agricultural Sciences, Yueyang 414000, China; 3Hunan Provincial Plant Protection and Quarantine Station, Changsha 410006, China; 4College of Plant Science and Technology, Huazhong Agricultural University, Wuhan 430070, China

**Keywords:** *Magnaporthe oryzae*, population structure, races, pathogenicity, resistance frequency

## Abstract

Studies on the population structure and variation of *Magnaporthe oryzae* in fields are of great significance for the control of rice blast disease. In this study, a total of 462 isolates isolated from different areas of Hunan Province in 2016 and 2018 were analyzed for their population structure and variation tendency. The results showed that from 2016 to 2018, the concentration of fungal races of *M. oryzae* increased and the diversity decreased; furthermore, 218 isolates in 2016 belonged to ZA, ZB, ZC, ZE, ZF and ZG, with a total of 6 groups and 29 races, in which the dominant-population ZB group accounted for 66.2%; meanwhile, in 2018, 244 isolates were classified into 4 groups and 21 races, including ZA, ZB, ZC and ZG, in which the dominant-population ZB group accounted for 72.54%. In 2018, isolates of ZD, ZE and ZF populations were absent, and the number of total races and isolates of the ZA and ZC groups decreased. Fungal pathogenicity was identified, with 24 monogenic lines (MLs) carrying 24 major *R* genes. The resistance frequency of *R* genes to fungal isolates in 2018 decreased significantly, in which except *Pikm* was 64.5%, the other monogenic lines were less than 50%. Rep-PCR analysis for isolates of Guidong in Hunan also showed that fungal diversity decreased gradually. The influence of *R* genes on fungal variation was analyzed. The pathogenicity of isolates purified from Xiangwanxian 11 planted with monogenic lines was significantly more enhanced than those without monogenic lines. All the results indicated that in recent years, the fungal abundance in Hunan has decreased while fungal pathogenicity has increased significantly. This study will greatly benefit rice-resistance breeding and the control of rice blast disease in Hunan Province.

## 1. Introduction

Rice blast caused by *Magnaporthe oryzae* can lead to a serious loss in rice yield and quality on a worldwide scale. As the most economical and effective measure, the application of resistant varieties has been widely used to control rice blast disease [1]. Rice resistance is based on the interaction between the rice resistance gene (*R*) and relative fungal avirulence gene (*AVR*) [2,3]. Pathogens invade plants in various ways, and meanwhile, plants try their best to defend against the invasion of pathogens through related resistant mechanisms [4,5]. The population of *M. oryzae* in rice fields is richly diverse, and it possesses frequent variations to overcome rice resistance. The continuous wide promotion of one resistant variety can cause the rice blast fungus to evolve into new pathogenic types and can lead to the loss of rice resistance [6]; meanwhile, this will have a great impact on the variation of the population structure of rice blast fungus.

Hunan is at the forefront of rice production in China. However, in Hunan, lots of rice planting districts are located in mountains and hills, and furthermore, the continental subtropical monsoon climate often leads to plentiful rainfall and a high humidity in the nature environment. The climate and geographical environment are suitable for rice blast occurrence and spread and result in the threat of blast disease and a serious loss of rice yield every year (http://www.hnppi.cn/web/hnzbzj/9803/9804/9807/9808/content_335136.html (last accessed on 11 January 2023)) [7]. On the other hand, the abundant ecological rice planting districts and varieties in Hunan may force rice blast fungus to be more diverse than in other provinces [8]. The improvement of the resistance of rice varieties to prevent rice blast disease has been a common measure to assure rice-production security. The consistent monitoring of the fungal population’s structure and pathogenicity constitutes the fundamental research for the control and forecast of rice blast [9,10].

Race classification based on Chinese race differential varieties and pathogenicity assays on monogenic lines (MLs) carrying major *R* genes have been widely used to analyze the fungal population. With these research methods, it was found that isolates from Jiangsu and Liaoning were similar in terms of both the pathogenicity and genetic structures [11], and *Pi9* and *Piz5* were the most effective *R* genes for isolates in Heilongjiang province [12]. Researchers also analyzed the dominant races of rice blast fungus in Fujian and identified the resistant rice varieties through a combination of indoor seedling inoculation identification and molecular marker detection. Furthermore, they guided the promotion and distribution of varieties [13]. The molecular identity and pathogenicity of rice blast fungus in 2012 in Hunan have been characterized, and *Pi9*, *Piz5*, *Pikh*, and *Pikm* were found to be effective blast resistance genes in Hunan province [8]. These studies have set up a great foundation for the demonstration of the genetic background of rice blast fungus in different provinces of China.

As we know, the nature environment and rice resistance genes are fatal impact factors to fungal variation. In former research, weather conditions have shown obvious discrepancies among three different years in Liuyang district, and analysis on isolates of each year showed that both the fungal population structure and pathogenicity varied [14]. Following the application of abundant new rice varieties and the change of the local climate condition in Hunan, the variation of rice blast fungus in fields is not clear. In order to monitor the genetic variation of *M. oryzae* and early warnings of blast disease, isolates collected from various regions of Hunan Province in 2016 and 2018 were analyzed. The objectives of this study were to determine (i) the fungal population structure and distribution as well as the variation trend, (ii) the pathogenicity and variation, and (iii) the influence of *R* genes to fungal variation.

## 2. Material and Methods

### 2.1. Fungal Isolates and Rice Varieties

In this study, 218 isolates in 2016 and 244 isolates in 2018 collected from different ecological regions in Hunan Province were analyzed (Appendix A). All dried panicle blast samples were sterilized with 70% ethyl alcohol for 10 s and rinsed with distilled water three times, and then immersed in distilled water for 2 h. The damp samples continued to lay on plastic plates with humidity at 27 °C for 24 h. Single spores were isolated under the microscope, then cultured on oatmeal agar, and finally stored on sterilized filter paper at −20 °C [8]. Regarding the name of stored isolates, “HN” represents Hunan, followed by the two digits of the year in which the samples were collected, and the left ones were the serial numbers.

The Chinese race differential varieties Tetep, Zhenlong 13, sifeng 43, Dongnong 363, Guandong 51, Hejiang 18 and Lijiangxintuan Heigu (LTH) were used for the classification of races [15]. Rice monogenic lines (MLs) were transferred into a single resistant gene with the background of LTH [16]. With LTH as negative control, 24 rice monogenic lines (MLs), respectively carrying *R* genes *Pik*, *Piz*, *Pi11*, *Piz5*, *Pikh*, *Pi7*, *Pii*, *Piks*, *Pi20*, *Pikp*, *Pia*, *Pi3*, *Pizt*, *Pit*, *Pita*, *Pib*, *Pish*, *Pi1*, *Pi5*, *Pi9*, *Pi12*, *Pi19*, *Pikm*, and *Pita2,* were used for the pathogenicity assay and resistance analysis of each major blast *R* gene [16]. In a greenhouse, ten seeds of each rice variety were planted and grown to the three- to four-leaf stage for pathogenicity assays.

### 2.2. Pathogenicity Assays

For pathogenicity assays, blast isolates were cultured on an oatmeal agar plate for 7 d with dark and white florescent light at 26 °C. Fungal spores were gathered with a 0.25% gelatin solution and filtrated with six layers of cheesecloth. The spore concentration was determined with a hemocytometer and adjusted to 2 × 10^5^/mL for the final inoculation.

During inoculation, a 20 mL spore suspension was sprayed on about 350 rice seedlings at the three- to four-leaf stage. Inoculated plants were transferred into chambers with 95% humidity without light at 26 °C for 24 h and then kept with light for the next six days; then, the disease level was evaluated with a 0-to-5 scale rating system, where 0 to 2 indicated resistance (R) and 3 to 5 indicated susceptibility (S) [17]. Each pathogenicity assay was repeated three times to confirm the disease reactions. The higher disease level was used if ratings disagreed.

The race identity was determined on the basis of disease reactions on the Chinese differential cultivars [15]. A total of 462 isolates were classified into six groups: ZA, ZB, ZC, ZE, ZF, and ZG, in which Z represents Zhongguo (China), and A to G were the abbreviations for each Chinese differential line. In case A was susceptible, the isolate was classified as ZA group; if A was resistant and B was susceptible, then the isolate was classified as ZB group; a similar rule was compiled for the other groups. Pathogenicity assays were repeated once to confirm the disease reactions.

### 2.3. Fungal DNA Extraction and Rep-PCR Detection

For fungal genomic DNA extraction, the filter paper with purified fungus was cultured on oatmeal agar for 5~7 days, and mycelium blocks with a diameter of 0.5 cm were transferred to 50 mL liquid complete medium (CM) to be cultured in table concentrator at 28 °C and 160 RPM for 3 days in dark. The mycelium collected through a gauze filter was dried with filter paper and ground into powder with liquid nitrogen. The genomic DNA was extracted according to the method provided by the DNA fungus extraction kit (OMEGA Fungal Genomic Extraction Kit).

Rep-PCR primers (POT2-1 and POT2-2) were designed through the end reverse repeat sequence of Pot2 transposon as described by George et al. [18]. The primers were synthesized by Changsha Qingke Biotechnology Co., Ltd., (Changsha, China) Rep-PCR amplification products were detected by electrophoresis with 0.5% Agarose and 0.75% Synergel. Rep-PCR procedure was as follows: 1 cycle of 95 °C for 2.5 min; 3 cycles of 94 °C for 1 min, 62 °C for 1 min, and 65 °C for 10 min; 25 cycles of 94 °C for 30 s, 62 °C for 1 min, and 65 °C for 10 min; and 1 cycle of 65 °C for 15 min. Rep-PCR assay was repeated once to confirm the fungal genome patterns.

### 2.4. Effect of R Gene on Fungal Population

In order to analyze the effect of the resistance gene on the fungal population, *Pi9* and *Pizt* monogenic lines and susceptible control—Xiangwanxian 11—were planted together in Guidong County, Hunan Province. Three independent experimental plots with a seedling density of 40 rows × 40 columns and 30 cm × 20 cm raw spacing were set up, respectively. A (control group): Xiangwanxian 11 was planted alone; B (*Pi9* test group): Xiangwanxian 11 was planted in the middle of a plot with 5 rows × 20 columns surrounded with *Pi9* monogenic line; C (*Pizt* test group): Xiangwanxian 11 was planted in the middle a plot with 5 rows × 20 columns surrounded with *Pizt* monogenic line. Seeds of rice materials were sowed at the end of May, and blast disease on rice were checked from the end of August. Samples of panicle blast of Xiangwanxian 11 from each plot were collected and purified with a single spore. For isolates from each plot, a race identification and pathogenicity assay on MLs were carried out to analyze the effect of *Pi9* and *Pizt* on the fungal population.

## 3. Results

### 3.1. Race Identity

In order to understand the population structure of the isolates, fungal races were classified under seven Chinese differential varieties. The isolates in 2016 were classified into 29 races and 6 groups (ZA, ZB, ZC, ZE, ZF and ZG), in which ZB group was dominant, accounting for 65.6%, and ZB13 was the dominant race, accounting for 39.91%, followed by ZC15 with 10.55% and ZB15 with 9.63%, respectively (Table 1); isolates in 2018 were classified into 21 races and 4 groups (ZA, ZB, ZC and ZG), in which ZB group was dominant, accounting for 72.54%, and ZB13 was the dominant race, accounting for 45.49%, followed by ZA5 with 10.25%. The major variation in these two years was performed on the subdominant race, which changed from ZC15 in 2016 to ZA5 in 2018, and meanwhile, the category numbers of the fungal race and group also decreased from 2016 to 2018.

### 3.2. Resistance Analysis of MLs

The results of pathogenicity determination on MLs are shown in Appendix A. For the isolates in 2016, the resistance frequencies of *Pikh*, *Pizt*, *Pik*, *Pi1*, *Pita2*, *Pi9*, *Piz5* and *Pikm* were over 50%, in which *Pikm* was the most effective, accounting for 80.75%. However, for isolates in 2018, even though *Pikm* was still the most effective, accounting for 64.5%, all the other MLs were less than 50%, and also more than half of the MLs were below 10% (Figure 1). The resistance frequency of different MLs varied greatly in recent years, especially the former highly resistant MLs. The resistance frequency of *Pi9* dropped from 68.08% in 2016 to only 13.5% in 2018, and also *Piz5* and *Pizt* were similar.

### 3.3. Fungal Genetic Diversity with Rep-PCR

Genome analysis of 27 isolates in 2016 and 73 isolates in 2018 in Guidong County was carried out with Rep-PCR, and 15 Rep-PCR patterns marked as A to O were found in total (Figure 2), in which A type was dominant and accounted for 51%.

For isolates in 2016, 9 Rep-PCR patterns, A, B, D, E, F, G, K, L and M, were found, in which the frequencies of F, G B, D, A, L and M were 18.52%, 18.52%, 14.8%,14.8%, 11.11%, 7.41% and 7.41%, respectively; both E and K patterns have only one isolate. For isolates in 2018, 8 patterns, A, C, F, H, I, J, N and O, were found, in which pattern A was absolutely dominant, accounting for 65.75%, followed by the J-banding type with 16.44%, while the other patterns were less than 5% (Table 2). The even distribution of different patterns for isolates in 2016 and highly intense distribution for isolates in 2018 suggested an obvious variation in the fungal genome and a decrease of fungal diversity in recent years.

### 3.4. Variation in Fungal Population around Pizt and Pi9 MLs

Panicle blast samples were collected from susceptible variety Xiangwanxian 11 in different test groups, and each isolate was purified with a single spore. There were 21 isolates in group A (susceptible control group), 23 isolates in group B (*Pi9* test group) and 40 isolates in group C (*Pizt* test group), respectively.

Race classification results showed that in group A, six races were found, in which ZB5 was dominant, accounting for 42.86%; in group B, three races were found, in which ZA1 was dominant, accounting for 47.83%; in group C, four races were found, in which ZA1 was dominant, accounting for 42.5% (Figure 3). These results showed that under the condition of the existence of MLs, the dominant race changed from ZB5 to ZA1, and the race number also decreased.

The pathogenicity assay on 24 MLs showed that there were also significant differences between the control group and test groups. For control group A, the resistance frequencies of MLs with *Piz5*, *Pizt*, *Pi9*, *Pikm*, *Pi20* and *Pita2* were 33.3%, 52.3%, 38%, 42.8%, 28.5% and 38%, respectively; however, for test groups B and C, the resistance frequency of all MLs was no more than 10% (Figure 4). The results suggested that the virulence of isolates from the test group was much stronger than that from the control group.

## 4. Discussion

In general, the interaction between *M. oryzae* and rice in nature maintains a dynamic equilibrium. In order to break through the defense of rice resistance genes, rice blast fungus will undergo different kinds of mutations, such as gene deletion, nucleotide substitution and transposon insertion [19,20]. Consistent monitoring of the rice blast fungus population in the field can clarify the distribution of prevalent isolates in different ecological areas and contribute to early warning, and it is also important for the rational application of rice varieties in order to control rice blast disease [21]. In this study, we identified the variation of the population structure and pathogenicity of rice blast fungus in Hunan Province in recent years and the effects of *R* gene on fungal variation. These results suggested that rice blast fungus went through significant changes in pathogenicity and genetic identity, and it will lead to a huge risk for rice production safety in the future.

The population structure of *M. oryzae* varied in different ecological regions and showed a dynamic development in different years. Fungal races and population analysis based on the Chinese differential system has been widely used [22,23]. In Hunan, 182 isolates in 2012 were found to belong to 6 groups and 28 races, and the dominant race had changed from ZG1 to ZB13 [8]. The consistent analysis on isolates in 2016 and 2018 showed that the dominant race and group was still the ZB13 and ZB groups, but the fungal population composition varied; particularly in 2018 it showed a significant increase of ZA group and decrease of ZC group, and the numbers of fungal group categories also diminished. The whole fungal population in 2018 got more concentrated, and the diversity was reduced. Meanwhile, the genetic background of the fungal population was also analyzed. The genetic polymorphism of *M.oryzae* is an important indicator for fungal population diversity. Different molecular marker methods, such as the restriction fragment length polymorphism (RFLP) method [24], randomly amplified polymorphic DNA (RAPD) technology, simple repeat sequence (SSR) labeling [25], Rep-PCR technology [17], and amplified fragment length polymorphism (AFLP), were widely used. In this study, Rep-PCR analysis was performed on isolates from Guidong County in 2016 and 2018. In 2016, nine types of Rep-PCR patterns were found and showed a uniform distribution, in which the highest proportion was 18.52%; in 2018, eight types were found, in which pattern A was dominant, accounting for 65.75%, and the other patterns were less than 17%. The isolates with the same genetic background got more concentrated from 2016 to 2018, and this was consistent with the conclusion on variations of race and population structure in this study. The change in fungal pathogenicity performance was direct evidence of fungal variation. For isolates in 2012, the resistance frequency of *Pi9*, *Piz5*, *Pikh* and *Pikm* was over 85% [8]. Even though most of these four major genes kept a relatively high resistance in 2016, in 2018, only the resistance frequency of *Pikm* exceeded 60%, and the other three *R* genes showed a precipitous decline. These results suggested that rice blast fungi in Hunan have undergone a distinct alteration and also provided a fatal warning for rice breeders.

The natural environment and resistant (*R*) genes in rice varieties with a large planting scale are the important factors affecting the population variation of *M. oryzae* [14,26]. There was no significant change for climates in Guidong County between 2016 and 2018. Therefore, *R* genes will be the core effect factor. In order to confirm the effect of *R* gene, isolates from a susceptible variety (LTH) with and without *Pizt* or *Pi9*-ML planted in high blast risk regions in Guidong were analyzed, respectively. The isolates from LTH alone performed with more diversity than those from LTH surrounded with *R* gene MLs. These results proved that *R* genes could affect the variation of the fungal population. Furthermore, the resistance frequencies of *Piz5*, *Pizt*, *Pi9*, *Pikm*, *Pi20* and *Pita2* to isolates from LTH alone ranged from 28.5% to 52.3%, but for isolates from the *Pi9* or *Pizt* test groups, they were no higher than 10%. This phenomenon is not only found in rice blast disease but also in wheat stripe rust [27].

Following the attention that has been focused on the damage from rice blast fungus, blast resistance has been the critical index for the authorization of a new rice variety. Hence, many resistant varieties have been promoted to farmers. Huazhan containing the major resistant gene *Pi2* proved to be an excellent resistant parent resource for hybrid rice breeding [28]. In recent years, the distribution data of main rice varieties in Hunan has shown that Huazhan series rice varieties accounted for more than 12% of middle and late rice from 2013 to 2017 [29], and the approved varieties of the Huazhan series also showed a gradual increase. As we know, the natural environment and resistant varieties were the major factors to induce fungal variation. In recent years, there was no drastic climate change in Hunan. Hence, the present situation in rice production may explain the variation in blast fungus. It is known that the promotion and planting of diversified rice varieties can help prevent and control the occurrence of rice blast [30]. Different control measures should be considered in response to fugal variation. In any case, we must pay great attention to the risk of the large-scale planting of varieties related to Huazhan, and we must also address the variation in blast fungus and breed new resistant varieties in advance.

## 5. Conclusions

As we all know, monitoring the structure and variation trend of rice blast fungus flora in rice planting areas is critical work for rice breeding. In the present study, we analyzed the fungal population structure and variation tendency of 462 isolates from different districts of Hunan Province in 2016 and 2018, showed the fungal variation through race population, genetic background, pathogenicity, and identified the influence of *R* genes on the fungal population. The results showed that from 2016 to 2018, rice blast fungus got more concentrated and its diversity decreased; additionally, the resistance frequency of *R* genes to fungal isolates decreased significantly in 2018, and fungus showed a stronger pathogenicity. In addition, Rep-PCR analysis of isolates from Guidong district in Hunan also showed that fungal diversity gradually decreased. Analysis on the influence of *R* genes on fungal variation showed that under the influence of MLs with *R* gene, the fungal dominant race varied and race number was reduced and also that fungal pathogenicity got stronger. This research is tightly related to the actual rice production situation, provides critical guidance for future rice breeding and contributes to the early warning of the potential risk of blast disease.

## Figures and Tables

**Figure 1 jof-09-00776-f001:**
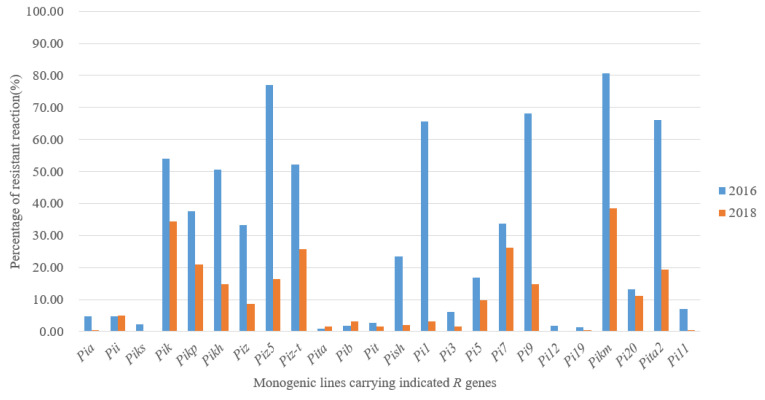
The resistance frequency of MLs to *Magnaporthe oryzae* in 2016 and 2018 in Hunan.

**Figure 2 jof-09-00776-f002:**
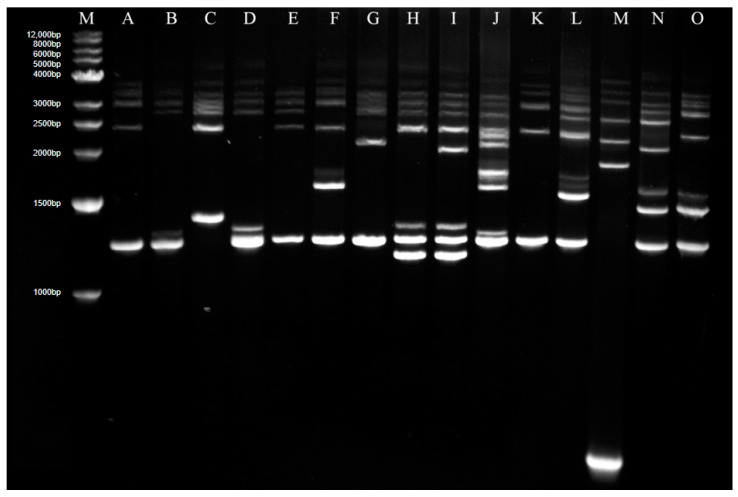
Rep-PCR analysis for genomes of all isolates in Guidong County. M—Marker; A–O—Different Rep-PCR patterns.

**Figure 3 jof-09-00776-f003:**
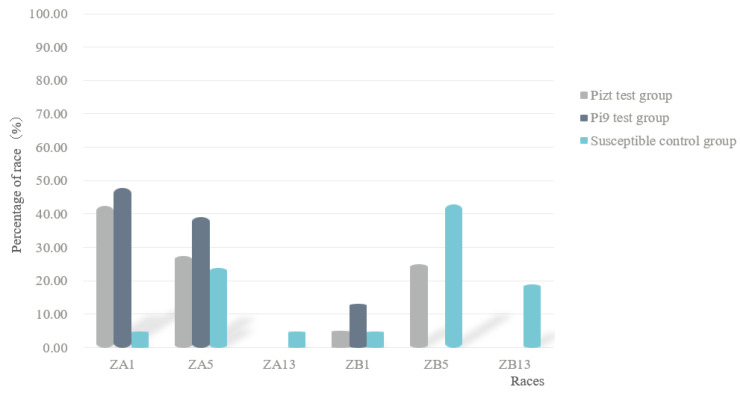
Fungal race structures of control group and of *Pizt* and *Pi9* test groups.

**Figure 4 jof-09-00776-f004:**
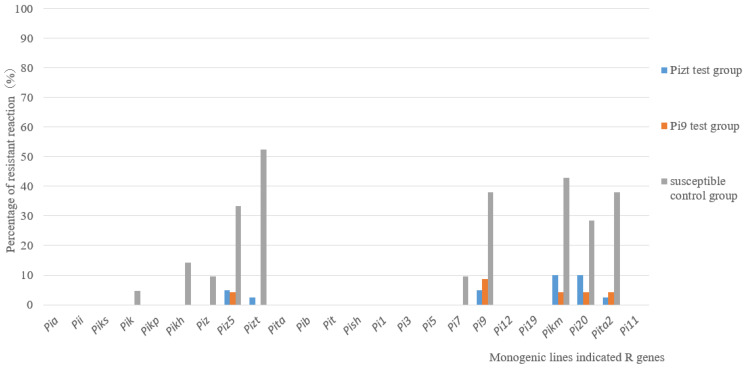
Pathogenicity assays on MLs for isolates from control group and from *Pizt* and *Pi9* test groups.

**Table 1 jof-09-00776-t001:** Race composition of *Magnaporthe oryzae* in 2016 and 2018.

	Races	ZA											ZB													ZC							ZE	ZF	ZG	Number of Race	Total
Year		ZA1	ZA3	ZA5	ZA9	ZA13	ZA15	ZA29	ZA31	ZA41	ZA45	ZA47	ZB1	ZB3	ZB5	ZB7	ZB9	ZB11	ZB13	ZB14	ZB15	ZB21	ZB23	ZB29	ZB31	ZC1	ZC5	ZC7	ZC9	ZC11	ZC13	ZC15	ZE4	ZF1	ZG1		
2016		4	3	2	0	9	1	1	0	1	1	1	3	1	11	0	6	1	87	2	21	1	1	5	4	0	2	2	1	0	13	23	1	2	8	29	218
2018		16	2	25	4	10	3	0	1	0	0	0	12	2	19	2	13	4	111	0	11	0	0	1	2	1	0	0	0	1	0	2	0	0	2	21	244

**Table 2 jof-09-00776-t002:** Isolate numbers of different Rep-PCR patterns.

Years	Patterns															Total
	A	B	C	D	E	F	G	H	I	J	K	L	M	N	O	
2016	3	4	0	4	1	5	5	0	0	0	1	2	2	0	0	27
2018	48	0	1	0	0	1	0	6	1	12	0	0	0	3	1	73
Total	51	4	1	4	1	6	5	6	1	12	1	2	2	3	1	100

## Data Availability

Not applicable.

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
