# Peer review of "Genetic Variation of Magnaporthe oryzae Population in Hunan Province"

_jof, 2023, doi:10.3390/jof9070776_

Round 1

Reviewer 1 Report (Previous Reviewer 2)

All responses are accepted. Good luck with paper!

Author Response

Thank you!

Reviewer 2 Report (New Reviewer)

The manuscript topic related to the investigation of genetic variation of Magnaporthe oryzae population in Hunan Province is suitable for the Journal of Fungi. The study is focused on determination of the population structure and distribution of the fungal pathogen, variation trend in the period of 2016-2018, pathogenicity and influence of R genes to fungal variation.

All parts of the manuscript are presented subsequently. The Abstract is focused on the main results and the Introduction is informative. All sections in Material and Methods are described in detail. Results are presented properly, all figures and tables, as well as supporting materials are clear and informative. The data combines information about a total number of 462 isolates from different districts of Hunan Province, fungal population structure and variation tendency. All the results indicate that the fungal abundance in investigated region has decreased, while fungal pathogenicity has increased significantly in recent years. The discussion is relevant to the results and is focused on the fungal variation with race population, genetic background, pathogenicity and the influence of R genes to fungal population. The Conclusions summarize the obtained results and highlight the contributions of the research that is related to actual rice production situation and is important for the control of rice blast disease in Hunan Province.

References correspond to the topic of the manuscript and are up-to-date.

A few inaccuracies in the text that need correction or clarification are listed below:

In the Abstract - The group ZD is pointed as missing in 2018, but it is not mentioned in 2016 at all. Why?

Page 2, row 88 - Is this a correct name of the varieties “s ifeng 43”?

Page 2, row 91 - “Tsunematsu et al. 2000” is mentioned with a full name, not as a number (16) from the References.

Page 3, row 110 - There is a description of the classification of six groups ZA, ZB, ZC, ZD, ZF, ZG, but in the Abstract different six groups are mentioned (ZA, ZB, ZC, ZE, ZF and ZG). This needs to be clarified.

Author Response

Dear Editor:

Thank you for constructive comments that have greatly improved our manuscript during this revision. We have provided here a “point-to-point” response to address these comments. The corresponding modifications in the revised text have been marked in yellow highlight. Responses to questions are marked in red text below

Comments:

In the Abstract - The group ZD is pointed as missing in 2018, but it is not mentioned in 2016 at all. Why?

Response:In general, there are few years where the flora can have isolates from all groups, and the 2018 isolates are only divided into four groups, without the three groups of ZD, ZE and ZF, which only emphasizes that the isolates in this year are relatively concentrated. In 2016, isolates can be divided into 6 groups, which is already abundant, and there is no special emphasis on which group of isolates are missing.

Page 2, row 88 - Is this a correct name of the varieties “s ifeng 43”?

Response:There is an extra space, modified to “sifeng 43” in text.

Page 2, row 91 - “Tsunematsu et al. 2000” is mentioned with a full name, not as a number (16) from the References.

Response: Modified in text.

Page 3, row 110 - There is a description of the classification of six groups ZA, ZB, ZC, ZD, ZF, ZG, but in the Abstract different six groups are mentioned (ZA, ZB, ZC, ZE, ZF and ZG). This needs to be clarified.

Response: Modified in text.

This manuscript is a resubmission of an earlier submission. The following is a list of the peer review reports and author responses from that submission.

Round 1

Reviewer 1 Report

It is a bit difficult to understand authors' train of thought in this manuscript. Some parts, even in the abstract, just does not make sense to me. For example, how did the authors reach the conclusion that fungal abundance in Hunan has decreased? Fungal race concentration increased, what does that mean? The authors also claimed there were no climate change in Hunan in 2016-2018. However, there was no reference to back it up. Overall, this study is focused on one specific geographic location and two specific years. The authors failed to highlight how it is of interest to the wider research community.

Reviewer 2 Report

In this manuscript, Zhirong Peng et al studied the structure and variation of Magnaporthe  in Hunan Province. Such a study is important for monitoring plant pathogens, which can lead to serious loss in rice yield and quality in worldwide. After reading the manuscript, I found there were several issues regarding to analyses, and results interpretation. I listed some of my concerns, hope this could help to improve the manuscript.

1)      A different number of isolates were analyzed for virulence and tested by PCR. At least the samples (in 2016 and 2018) should be normalized. You cannot interpret the result of your PCR test the way you did: you took a different number of samples. And here's the question: those isolates that you tried using PCR - have they been analyzed for virulence? Where is the correlation analysis of these two approaches? If you do not have it, it is better not to mention the results of PCR. They turned out to be dependent on artificial selection of genotypes

 You have shown that the R genes had a major impact on the structure and dynamics of the population. Good. Please provide information about the dynamics of the realizing varieties in Guidong County  between 2016 and 2018 to confirm your statement.